# Interrogating the Role and Value of Cultural Expertise in Law

**John R. Campbell**

Department of Anthropology and Sociology, School of Oriental and African Studies, London WC1H OXG, UK; jc58@soas.ac.uk

**Abstract:** It is common for litigation to draw upon expert evidence to assist a judge to arrive at a balanced decision. This paper examines the role of one type of expert evidence submitted to courts, namely cultural expertise (CE), which provides information on socio-cultural issues such as kinship, family, marriage, customs, language, religion, witchcraft and so on. This type of evidence is primarily the result of qualitative, ethnographic research. I begin by examining the views of experts who have provided CE to courts/mediators; I then look at how judges view and make use of CE, and finally I examine lawyers' views on CE. To address gaps in published research, I interviewed British barristers to understand how they make use of experts in the cases they litigate. Finally, I have surveyed legal decisions made by all British appellate courts to arrive at an approximate idea of the extent to which CE has been submitted in English and Welsh courts. I conclude that the extent to which CE—and other types of socio-legal evidence—is submitted varies considerably depending upon the legal/evidentiary procedures followed in different jurisdictions and in different countries.

**Keywords:** cultural expertise; expert evidence; judges; barristers; appellate courts

## 1. Introduction

Scholarship on the relation between anthropology and law has slowly recognized that characterizations of the two disciplines as being epistemologically incommensurable are unhelpful and unproductive. Anthropology seeks to address social life in all its variety. As anthropologists have argued, culture as a system of values and meaning is unbounded, dynamic and involves a process of redefining and contesting social norms and forms of power. Law, on the other hand, is a process by which those in power attempt to regulate conflict by deciding/adjudicating the limits of acceptable —restated as enforceable 'legal'—behavior and social norms. In this view, anthropology and law are mutually constitutive, even though many legal professionals perceive their role as being neutral, objective, and independent of cultural concerns and many anthropologists argue that anthropological evidence on cultural issues is routinely rejected by the law.

As the legal process has engaged with socio-cultural and technological change it has had to deal with a more socially diverse range of litigants/claimants, the law has increasingly accepted evidence from a growing range of 'experts'—from the social, cultural, behavioral, scientific, technical, judicial and other fields—who have provided extensive information about the impact of change on social and cultural life. Arguably, in grappling with social change, law has been forced to address the ubiquity of culture as it decides the rights and wrongs of individuals from diverse cultures, communities, and states.

In this paper, I assess the emerging role of 'cultural expertise' or CE in judicial proceedings in western Europe and North America, a process which has long antecedents but which has recently been spurred by the work of Livia Holden (2011, 2020) and her European Research Council funded project

'Cultural Expertise in Europe—What is it useful for?'[1]. The research which Holden has undertaken and generated has focused primarily on the contribution of academic researchers who have provided expert evidence to courts and decision-makers in an attempt to assist judges/mediators to better understand the cultural issues which arise in different types of disputes.

Holden's initial definition of CE reflected her focus on the role of anthropologists who, she argued, were in a position to provide specialized knowledge to a range of decision-makers who decide disputes in which 'culture' was an issue (Holden 2011). More recently she (Holden 2020, p. 45) has adopted a wider definition of CE as 'the special knowledge that enables socio-legal scholars, experts in laws and culture, or, more generally speaking—the so called culture brokers—to locate and describe relevant facts in light of the particular background of the claimants and litigants and for the decision-making authorities'.

Holden's project raises a number of interesting issues which I address in this paper. First, I review published research that focuses on the perspective of the 'experts' who have submitted CE[2] (given the wide range of experts and differences in their disciplinary training, it is not useful to treat them as a homogenous group). However, there is limited research regarding the value that these experts bring to judicial decision-making. Not to put too fine a point on it, some lawyers and judges are concerned that expert witnesses should be regulated and controlled to prevent them from usurping the role of judges and juries from deciding the 'ultimate issue', i.e., the factual issues arising in the case (though in the U.S. and elsewhere the ultimate issue rule has been set aside).[3] Second, there is very little research about the perspective of lawyers and judges on the value of CE. Finally, research on CE is primarily concerned with individual cases. Existing research provides little insight into the socio-political or structural factors which give rise to different types of claims, be they asylum, criminal or civil claims, nor does it ask whether a legal victory provides a satisfactory resolution of the factors which gave rise to the case or whether it creates new forms of subordination to the state.

As studies of litigation have pointed out, it is useful to explore the legal process and issues surrounding the value of evidence from multiple points of view to better understand the legal field, the meanings which actors ascribe to their activities and the different contexts of dispute resolution (Munger 1990, pp. 601–2). This paper examines the role of CE in western Europe and North America. I begin by reviewing research on the work of experts who submit CE. In the second section, I examine the contribution of CE from the point of view of judges. In Section 3, and in order to situate CE in the wider legal, adversarial context, I look at how barristers who litigate cases in Britain make use of experts. Finally, I analyze data from a survey of cases reported on Bailii, a comprehensive legal database managed by the British and Irish Legal Information Institute, which suggests that CE is one relatively minor type of expert evidence submitted in British courts. The paper concludes by situating CE in the wider legal, adversarial context where it is relied upon primarily in asylum and immigration law, civil claims, and international criminal law, and secondarily in international human rights law and family law.

## 2. The Perspective of Cultural Experts

The earliest paper exploring the use of cultural experts was written by Clark (1953), who described the work of anthropologists and psychiatrists involved in civil litigation in the 1950s which overturned racial segregation laws in the U.S. Clark identified a number of difficulties facing experts involved in these cases and argued that 'it will be necessary for the professional societies among

---

1. See: https://culturalexpertise.net/ (accessed on 9 July 2020). My appreciation of CE has benefited greatly as a result of participating in this project as a research associate.
2. Holden has convened a number of seminars and conferences with experts, lawyers, and judges from across Europe which have been summarized but not analysed. I cite these problematic and somewhat contradictory 'data summaries' below.
3. See, for instance, rule 704 in the U.S. Federal Rules of Evidence at: https://www.law.cornell.edu/rules/fre/rule_704 (accessed on 15 September 2020).

the social and psychological sciences to develop safeguards against possible ethical abuses; e.g., flagrant manifestations of prejudice, distortion of data and deliberately misleading interpretations' (pp. 9–10).

Subsequently Rosen (1977) wrote about the involvement of anthropologists in litigation between 1950 and the early 1970s. Rosen discussed anthropological involvement in civil litigation in the U.S. which sought to overturn racial segregation and which supported First Nation peoples. He identified a number of landmark legal cases—including *Brown v The Board of Education*, *Plessy*, *Loving v Virginia*, *Wisconsin v Yoder* and *United States v State of Washington*—in which anthropological evidence played an important role in striking down segregation and anti-miscegenation laws and upholding the importance of culture to secure the rights of religious minorities and native land claims.[4]

Today, it is clear that anthropologists, historians, geographers and individuals from other disciplines have engaged with the law on a global scale with respect to legal claims made on behalf of, and sometimes against, individuals and social groups. With regard to the individual claims, anthropologists and others have provided expert evidence in cases involving crime (Fontein 2014), genocide (in the international criminal courts; Eltringham 2013; Anders 2014; Wilson 2015), in claims affecting different diasporas, i.e., South Asians (Holden 2011; Menski 2013); Africans (Clarke 2017) and in asylum and immigration proceedings in the U.K. (Good 2004, 2007, 2008; Hoehne 2016; J. R. Campbell 2017, 2020a), the U.S.A. (Berger et al. 2015; Ngin 2018) and Sweden (Rabo 2019). Anthropological evidence/CE is likely to be of value in asylum proceedings for two reasons. First, Immigration Judges (IJs) applying international humanitarian law are required to understand whether an asylum applicant has 'a well-founded fear' of persecution which requires them to assess an applicant's account of persecution and flight, i.e., his/her subjective and objective fears. Second, unlike courts in other jurisdictions[5], asylum courts are allowed to admit hearsay evidence as well as other evidence which an appellant may wish to submit (though it may attach little weight to this evidence). The focus of the work by experts has been to use anthropological/CE reports—which is uniformly viewed as hearsay evidence—to obtain recognition/protection for individuals.

The second focus of expert evidence relates to disputes concerning indigenous people's land claims. In these cases, anthropologists and historians have provided ethnographic and ethno-historical data in 'native'/indigenous land disputes and in related political struggles in Alaska and Canada (Feldman 1980; Ray 2011), in Australia (Weiner 1999) and in central and South America (Loperana et al. 2020; Hale 2020). However, in recent years, this type of litigation has seen a growing 'refusal' by First Nations/native Americans to engage with the law in its terms in an attempt to refuse political solutions imposed by the state (Hale 2020; Loperana et al. 2020). This paper focuses on litigation by or on behalf of individuals.

Most of the literature on expert witnessing is written primarily by anthropologists concerned with legal proceedings in which 'culture is teleologized in courts of law, by being treated as 'objective evidence" (Good 2008, p. S47). Some anthropologists have tended to distance themselves from lawyers and judges who are said to be 'empiricist and positivistic' and who 'are trained to think in radically different ways' than anthropologists and social scientists (Geertz 1983; Good 2004, p. 129; Bens 2016). However, as Loperana et al. (2020, pp. 588–89) have argued, this distinction obscures the fact that both anthropology and law emerged as powerful forms of knowledge during colonialism.

---

4　Gormley (1955) notes the involvement of anthropologists in 'Indian tribal claims' dating back to 1895 and possibly earlier. He cites '*Choctaws et al. v. United States, 34 C.Cls. 17, 54* et seq.'

5　In the UK, sec. 114 of *The Criminal Justice Act 2003* defines hearsay evidence as any 'statement not made in oral evidence in the proceedings.' Reliance on a statement made otherwise than while giving evidence to prove the truth of a fact asserted remains hearsay. Hearsay evidence is inadmissible and the rule applies: (a) to both examination in chief and cross-examination; (b) whether the statement was made by the witness personally or by some other person; (c) to any 'out of court' statement, whether oral, written or otherwise; and (d) to statements given as evidence of the truth of its contents—if the statement is given for any purpose which is relevant to the facts in issue in the case, it is admissible, for example, evidence given as to a person's state of mind, rather than what was actually said.

Furthermore, as Eltringham (2013) and Wilson (2015) have argued, an assessment of anthropological involvement in the legal process requires a clear understanding of the adversarial legal process. This begins with lawyers identifying and instructing experts in an attempt to win a case, and culminates when a judge, having assessed the evidence and legal arguments, arrives at a decision. Engaging with the law as an expert may reify and strengthen the law by reinforcing the role of the state over an individual or a community; though occasionally experts may provide evidence which successfully challenges and overturns case law.[6] (cf. Rosen 1977). It is important to step back from Clarke's (2020, p. 585) statement on the different tasks of the two disciplines when she states that: 'Where the court's purpose is to establish hegemonic order through judgement, one of anthropology's purposes is to illuminate alternative cosmologies and possibilities rendered by diverse subjectivities'. Experience with the legal process teaches us that the law only perceives a defendant/litigant as either guilty or not guilty; judges in criminal and other jurisdictions have little patience with arguments about culture/cultural relativity, witchcraft/sorcery or which seek to challenge the common sense reasoning they rely on (when this occurs, judges quickly act to rule such arguments inadmissible; (Fontein 2014; Anders 2014)). I argue that the specific role played by the two disciplines needs to be examined on a case by case basis.

Hoehne (2016, p. 253) has argued 'that it is not a fundamental epistemological divide, but rather massive power differentials that characterize the relationship between social anthropologists and legal practitioners' and that this difference 'sits uneasily with the professional, moral and ethical standards of the discipline'. Hoehne argues that anthropologists providing evidence in courts of law must adopt a 'strategic form' of essentialism, as opposed to a post-positivist position reflecting the contingent nature of anthropological knowledge if they 'are to fulfil the requirements of the legal process and maintain one's role as expert' (p. 257). Hoehne's point is echoed by H. Campbell et al. (2017, p. 333) who also make a case for 'strategic essentialism'—which the authors define as 'the pragmatic practice of defining cultural groups or their practices in ways that emphasize commonalities rather than differences while recognizing that a more expansive analysis would include greater nuance and complexity'[7]—via a well-informed understanding of immigration and criminal law. They argue that anthropologists should defend the rights of subaltern peoples caught up in powerful legal proceedings that may unfairly imprison, execute or deport them to their country of origin. The authors argue that anthropologists 'need to find a way to use our ethnographic expertise to not just defend individuals' caught up in legal proceedings 'but also leave a positive social and institutional [i.e., professional] record' (H. Campbell et al. 2017, p. 333). For these anthropologists, witnessing is a political act.

The issue of the legitimacy attached to an expert's evidence was raised by Rosen (1977, p. 555) who observed that most anthropologists 'may not understand how expert testimony fits together with judicial reasoning and legal precedent, and precisely how the court's investigation of the facts articulates with the form of knowledge he possesses.' In particular, and in relation to the rules governing testimony/evidence which allow 'expert witnesses' to testify about specific social and physical phenomena which they have not personally witnessed[8], Rosen draws attention to four issues which anthropologists—and by definition other types of expert—must take into account if their work is to be effective. First there are 'concerns about the adequacy, context, and form of presentation of anthropological evidence in an adversary proceedings' (p. 556). Is testimony adequate given the issues

---

[6]　See the civil cases identified by Rosen (1977) and *ST (Ethnic Eritrean—nationality—return) Ethiopia CG [2011] UKUT 00252(IAC)*, an asylum claim decided by the Upper Tribunal of the UK's Immigration and Asylum Chamber which overturned five precedents on the basis of expert/CE evidence.

[7]　As Clarke (2020, p. 587) has noted, the term 'strategic essentialism' was introduced by Gayatri Chakravorty Spivak to refer to 'a political tactic in which minority groups, nationalities, or ethnic groups mobilize on the basis of shared gendered, cultural, or political identities to represent themselves.' The contrast between how anthropologists describe minority groups in legal claims and how the latter describe themselves is important and is addressed by Hale (2020) and Loperana et al. (2020).

[8]　In the UK, experts are permitted to testify about relevant socio-cultural issues, culture, family, foreign law, and a growing range of new topics related to the rise of technology, for instance, video/surveillance film, DNA, drone technology and so on.

raised by a case? Will evidence be inappropriate or distorted by adversarial proceedings? What evidentiary standards are anthropologists required to meet? Second, there are questions regarding 'the mutual effect that courts and anthropologists have on one another' (p. 557). Have anthropological concepts been affected or appropriated by litigation? How has anthropological testimony been shaped by adversarial argument and judicial reasoning? How can anthropologists balance their work as experts against wider professional obligations? Third, Rosen notes that there are serious questions regarding 'anthropologists conceptions of their role in such proceedings, and how the courts and the profession may contribute to appropriate reforms' (p. 557). For instance, do anthropologists provide crucial information for judicial decisions or is their information 'simply useful for rationalizing judgements that are founded on other, perhaps judicially less palatable bases?' Rosen asks, 'How should anthropology approach the ethical implications of expert testimony?'

There are, therefore, important political and practical issues which can undermine the effectiveness of experts and the extent to which their testimony will be recognized by a court as constituting valid evidence (and be accepted by their profession) which relate to the jurisdiction in which a case is heard/tried and the epistemological approach followed by the expert. Two examples will have to suffice. While Immigration Judges (IJs) in the U.S. 'have broad discretion to conduct and control immigration proceedings and to admit and consider relevant and probative evidence, including witness testimony', their discretion is not without limit. In the '*Matter of J-G-T- Respondent*'[9] heard by the Executive Office for Immigration Review in 2020, the Board of Immigration Appeals decided that,

> . . . in assessing whether to admit the testimony of a witness as an expert, an Immigration Judge should consider whether it is sufficiently relevant and reliable for the expert to offer an informed opinion, and if it is admitted, the Immigration Judge should then consider how much weight the testimony should receive. If a party challenges the expert's qualifications, it is generally best to allow the party, upon request, to *voir dire* the witness before the testimony is presented in full. In considering how much weight to give an expert's testimony, the Immigration Judge should assess how probative and persuasive the testimony is regarding key issues in dispute for which the testimony is being offered. However, to the extent that the record contains contradictory evidence, the Immigration Judge should explain why inferences made by the expert are reasonable and more persuasive than the other evidence presented.

Second, and in relation to transitional justice cases which are heard by International Criminal Tribunals, Jones (2015) raises important questions about the status of expert evidence and how it is evaluated. She notes that different researchers and organizations involved in providing expert evidence to international tribunals follow different epistemological approaches, provide different 'versions of reality' and are able to secure varying levels of legitimacy. Jones argues that in transitional justice, 'first-hand experience' of the countries in question 'and more ethnographic and empirical understandings[10] were generally dismissed by the bench of international judges' in favor of the 'foreign expertise' provided by international NGOs whose evidence more closely resembled the judges own 'legal and fact-based approach' (p. 296). For Jones, the key question concerns the legitimacy which the courts attach to the versions of reality provided by different experts: she asks, whose voices—not only which experts but which local narratives—are being heard? In this regard, Wilson's (2015) review of how qualitative research faired in international criminal trials suggests that social scientists should not attempt to challenge the authority of the court and that reports should be written in clear, non-technical and jargon-free language. His analysis indicates that evidence provided by social

---

9   This case is cited as Matter of JGT-, 28 I&N Dec. 97 (BIA 2020: https://www.justice.gov/eoir/page/file/1319951/download).
10  The author explores the value of narrative interviews in providing relevant evidence, but there is no reason to limit research to this method.

scientists was more frequently cited in judicial decisions than that of military and police experts, document verification experts, financial experts, engineers, and medical experts (p. 732).

The final issue raised by Rosen is what he calls 'the cultural argument', though as discussed below, 'culture' is not the only issue which anthropological/CE experts are asked to address. As Rosen frames the issue, legal cases

> present problems of interpreting to the court the language and concepts of the party involved, and the relation between the legal issues posed and the relevance of anthropological findings. Working closely with counsel, anthropologists have also been instrumental in formulating highly creative arguments that may influence the course and result of a case (p. 567).

Rosen cites several successful 'creative arguments' which includes litigation about the use of peyote by Native Americans, snake-handling cults in the state of Tennessee, the issue of 'arctic hysteria' and 'witchcraft murders'. The creation of a 'cultural defence' in the U.S. arose later (see Dundes Renteln 2002, 2004). The issue of whether it is right that anthropologists should 'interpret' culture rather than allow litigants an opportunity to speak for themselves in legal proceedings is determined in part by whether the litigation strategy adopted by legal counsel empowers parties to speak for themselves and by the disciplinary training of the experts who submit evidence. In group legal claims, such as land right claims, there has been a shift away from the anthropologist acting as the 'expert' to one of facilitating local groups to speak for themselves (Loperana et al. 2020; Hale 2020).

The social sciences have still not addressed the professional and pragmatic issues raised by Clark and Rosen nearly forty years ago. A lack of preparedness for adversarial proceedings can result in shock and surprise when judges dismiss an experts' evidence or when the judiciary appropriates and 'misuses' anthropological concepts and evidence (Riles 2006).[11] Apart from a desire to be useful to the courts, most experts possess a limited awareness about how the courts 'judicialize culture'[12] and they are unaware that they may be required to disclose their sources of information.[13] While professional associations have engaged with the general ethical issues facing members of their discipline, their guidance has left individual members to negotiate thorny issues arising from multiple and cross-cutting obligations to one's informants, and responsibilities to funders, one's university or employer, the nation-state in which fieldwork was conducted, and the courts.[14] The general approach to ethics adopted by professional bodies has had two principle implications for those who engage in expert witnessing. First, relatively few university academics engage in applied research, including expert witnessing (J. R. Campbell 2020a). Second, the ethical approach adopted by those who provide CE reflects their individual understanding of the challenges they confront. While some anthropologists would agree that they have an obligation to support subaltern people confronted by the law (Holden 2019, p. 190-f), they do not necessarily believe that they should challenge the judiciary—for example, by disputing judicial reasoning or challenging legal arguments made by the state—or take a political stance with regard to witnessing.

The problems which arise when anthropologists attempt to raise 'the cultural argument' are complex. First, even in instances where the courts have accepted cultural evidence provided by

---

[11] It appears that only a small number of anthropologists who provide expert evidence are familiar with adversarial proceedings, namely Rosen (1977, 1989), Good (2007), H. Campbell et al. (2017), Dundes Renteln (2004) and J. R. Campbell (2015, 2017, 2020b).

[12] See (Rosen 1989; Good 2007; Howes 2005).

[13] See, for instance, the guidance set out by the UK's Crown Prosecution Service (2019a).

[14] See Manners (1956) for an informed and early discussion of the complex issues which arose when anthropologists provided expert evidence in Native American land claims. For the guidance provided by the American Anthropological Association see: https://www.americananthro.org/LearnAndTeach/Content.aspx?ItemNumber=22869&navItemNumber=652. The ethical code of the American Historical Association only notes that historians should avoid any 'conflicts of interest' (see: https://www.historians.org/jobs-and-professional-development/statements-standards-and-guidelines-of-the-discipline/statement-on-standards-of-professional-conduct#Reputation). The American Association of Sociology also sets out very general ethical guidelines, see: https://www.asanet.org/sites/default/files/asa_code_of_ethics-june2018.pdf (sites were accessed on 20 August 2020).

an anthropologist and the argument has been generalized for use across a range of cases—the best documented example is the concept of a 'cultural defense' employed in U.S. courts—the way in which cultural evidence is handled indicates that trial judges fail to apply a sufficiently robust test to assess the veracity of a 'cultural claim', because they rely upon their own 'common sense' to decide cases (Dundes Renteln 2005; Wilson 2015; Maucec 2020). In short, there is a significant difference between the way that anthropologists and judges understand and employ the notion of a cultural defense. It is also the case that the value of a legal victory may be more pyric than real depending on how the state implements the decision and whether a successful litigant is subordinated to the state. In addition, subsequent legal cases may challenge and undermine previous legal victories. This is certainly what occurs in civil and in asylum claims where the direction of the law is dependent upon who possesses the resources to use it. As Nader (Nader 2001–2002) has argued, in legal systems dominated by the state the plaintiff role tends to atrophy and, over time, the legal system disadvantages individual plaintiffs/claimants.

Secondly, 'cultural issues' may not be the principle or only issue raised in a legal case. This situation may arise as a result of the 'liberal' approach adopted by a court when it fails to understand, take cognizance of or reconcile conflicting concepts of culture and justice (Douglas 2005), when a court frames the issues narrowly giving short shrift to underlying cultural issues (Roy 2005), when a court seeks evidence about foreign law and practice (J. R. Campbell 2020a) or when judges exclude expert evidence (Currie 2005; Wilson 2015).

## 3. The Perspective of Judges on the Value of CE

Jones (1994) provides an excellent overview of the English judiciaries' attempt to regulate 'expert' witnesses in trial proceedings. She observed that when the judiciary realized that experts disagreed in the methods they adopted and the conclusions they provided to the court—even though disagreement was and remains an essential aspect of scientific enquiry—they saw experts as 'suffering from partisanship' (p. 97-f). The judicial view of science and of scientific experts was compounded when experts were called by both parties and/or when fees were paid to experts to provide evidence. This situation contributed to a 'distorted' view of science by the judiciary, namely that scientists should provide evidence that was impartial, disinterested, and neutral. However, when expert evidence deviated from the ideological construct created by the judiciary, judges used procedural rules to 'dismiss expert evidence as the weakest kind of testimony.'

The principle 'stick' devised by the judiciary to 'beat the expert witness' took the form of new rules of evidence and procedural/admissibility rules which 'forced experts to fit their evidence into an artefact of the law's design', namely a written report which addresses issues identified by lawyers. In most jurisdictions, experts have an 'overriding duty' to assist the court—which overrides their obligations to the party which instructed them—and their reports must conform to a specific format.[15] In the United Kingdom, for instance, experts must not advocate on behalf of the claimant.[16] The rules give the lawyer who submits the evidence maximum control over the expert and her evidence, and they give judges the power to admit or refuse expert evidence.[17] Furthermore, the adversarial system relies on the parties to cross examine experts in a process which seeks to undermine the weight that

---

[15] The UK Ministry of Justice sets out clear regulations governing the role of experts (see: https://www.justice.gov.uk/courts/procedure-rules/civil/standard-directions/general/experts; accessed on 18 August 2020). In the U.S. the case of Daubert has shaped Rule 702 of the Federal Rules of Evidence (see, for instance: https://www.law.cornell.edu/rules/fre/rule_702) and see the discussion of this by Golan (2008).

[16] Legal representatives may instruct an expert relatively late in the day which leaves the expert with limited time to undertake research and write a report. Many instructing solicitors fail to tell an expert about the Civil Procedure Rules which can result in the court rejecting their report.

[17] The UK's Crown Prosecution Service provide a slightly different view of experts. Their guidance states that 'The duty of an expert witness is to help the court to achieve the overriding objective by giving opinion, which is objective and unbiased, in relation to matters within their expertise' (my emphasis, Crown Prosecution Service 2019b).

will be placed on their evidence leaving it to the jury or the judge—depending on the jurisdiction in which the case is being heard—to assess the value of the proffered evidence (Golan 2008, p. 917-f). In effect, experts have been co-opted by the legal process.

The judiciary regulates expert witnesses in part through imposing evidentiary hurdles which vary between different jurisdictions and countries. For instance, the Australian judiciary rely upon 'the field of expertise rule' which has been expressed as 'whether the subject matter on which expert evidence is being adduced 'is such as to be the proper subject of expert testimony" or is a 'recognized field of specialist knowledge' (Redmayne 2001, p. 100). In the U.S., the Frye and Daubert 'tests' have been devised by the judiciary as an attempt to shift the focus of admissibility away from the credentials of the expert to the specific nature of the scientific knowledge s/he proposes to submit to the court. In this way the Federal courts created a new 'law of evidence' that was intended to prevent the expert from giving his or her opinion on the 'ultimate issue', i.e., the precise factual issues before the jury (Golan 2008, p. 921-f). All the different tests, however, including the hearsay doctrine which allowed judges to exclude 'opinion evidence', have proved unwieldy and have been applied inconsistently by judges. As Redmayne has argued, the tests 'are so flexible that they could be applied strictly or laxly, depending upon the court' to 'screen scientific evidence to ensure its reliability' (Redmayne 2001, pp. 104–5). Such rules exist in many countries and may be used to regulate and prevent expert evidence from being given. In this regard, Currie (2005, p. 81) reports a civil defamation case in Canada in which the trial judge decided that the expert evidence to be provided by a sociologist and an anthropologist about racism in the local police force was inadmissible because sociology and anthropology 'lack the precision and specificity which characterizes a science like chemistry or an area of technical expertise like engineering.'[18]

Admissibility is less likely to be an issue when a judge, rather than a jury, is deciding a case. This is particularly true with respect to hearing minor criminal offences, most civil claims (which are increasingly expected to be arbitrated/mediated) and asylum and immigration appeals.[19] While it is the case that in some countries there is a right to a trial by jury for serious offences and in important civil claims, the reality is that the number of jury trials for all types of claims have declined substantially in North America (Kritzer 2004; Galanter 2004) and the U.K. (Dingwall and Cloatre 2006).[20] It is important to add that across Europe, and in many other countries, civil claims are heard by judges and trial for criminal cases is rare (Leib 2007–2008). In France and Belgium, which follow the civil law tradition, only serious criminal offences are heard by a jury (Germain 2019).

Even though courts use evidence/admissibility standards, many lawyers and judges distrust experts who are seen as 'hired guns' because it is believed that their evidence can mislead juries and judges and may result in wrongly deciding cases (Mosteller 1989; Mantle and Chenane 2014; Wortly and Ward 2019). Judicial views of experts are not without some justification given the extent to which expert evidence is internationally marketed for all types of litigation[21] and because experts are appointed who possess diverse forms of disciplinary training and who have different motivations, including personal gain for providing their expertise.

It is useful to step away from a focus on the rules which judges can use to regulate expert evidence to understand the importance of CE in different countries. This task requires us to distinguish between proposals made to the judiciary which advocate the adoption of an 'appropriate cultural test' (cf.

---

[18]  On appeal the judge's decision was set aside.
[19]  For the UK see: Courts and Tribunals, 'Civil Justice in England and Wales' at: https://www.judiciary.uk/about-the-judiciary/the-justice-system/jurisdictions/civil-jurisdiction/#:~:text=The%20vast%20majority%20of%20civil,is%20%E2%80%93%20and%20then%20giving%20a; for Canada see: Government of Canada, Department of Justice 'The role of the public' at: https://sistemasjudiciales.org/wp-content/uploads/2018/03/947.pdf; for the U.S. see: the Department of Justice guidance on '201. Jury trials in civil cases' at: https://www.justice.gov/jm/civil-resource-manual-201-jury-trials-civil-cases and the EOIR IJ Bench Book Tools, Guide, Evidence Guide at: https://www.justice.gov/eoir/page/file/988046/download.
[20]  My survey of the literature did not find any indication that either a judge's directions to a jury or jury selection/screening played a role in the outcome of the hearings/trials discussed in this paper.
[21]  See, for instance, 'Expert Evidence International Ltd.' at: https://expert-evidence.com/ (accessed on 1 September 2020).

Ruggiu 2019) from research which explores how judges actually assess and make use of CE. There are relatively few published studies which explore the second issue.

Cooke's (2019) study of Finland confirms that while it is possible for the courts to accept CE, this has not happened (though why this does not occur is not discussed). Instead she focuses on the role of 'informal' expertise which occurs when community members are appointed by a judge to address cultural issues. Lopes et al. (2019) 'survey' of Portuguese law identifies and discusses a small number of legal cases relating to migrants whose culture may be relevant when they are charged with specific crimes, i.e., Female Genital Mutilation, forced marriage, specific types of sexual acts, placing children for adoption, the marriage of minors and a child's right to education. In Portugal, expert evidence is only introduced 'through the initiative of a judge' which rarely occurs. The authors found that while 'the courts were relatively indifferent to cultural factors', an extensive network of 'mediators' operated at the community level 'who were mobilized to defend ethnic minority rights' (p. 66).

The situation in Italy is slightly better known. For example, Holden's survey of judges, lawyers and experts suggests a fairly small number of experts are appointed and that experts provide evidence primarily in immigration law (44%), refugee and asylum law (22%), family law (19%) and civil law (16%).[22] Ciccozzi and Decarli (2019) state that while a wide range of experts are recognized in Italian law, 'in most cases, judicial actors are not aware of the cultural complexities of the groups involved in trials and have no on-site experience with the populations to which the parties belong' (p. 37). Furthermore 'the number of experts appointed in Italian trials has been very low'. The authors provide a detailed discussion of Ciccozi's involvement as an anthropological expert in the 'L'Aquila trial' which involved an assessment of whether expert scientific evidence concerning the risk of seismic activities was understood (and ignored) by residents of L'Aquila village with fatal consequences.

With regard to the situation in Sweden, Holden's survey of judges, lawyers and experts[23] found that CE was primarily relied upon in immigration law (13%), refugee and asylum law (22%), international human rights law (11%) and in family law (10%). The majority of judges and lawyers had used experts in less than 10 cases. Rabo (2019) sets out a more nuanced history of CE in Swedish law in which she examines cases concerned with the land rights of the indigenous Sami people, cases of discrimination against Roma people and cases—concerned with deportation, adoption, hate speech, divorce, spirit possession, FGM and murder—involving 'new immigrants'. She notes that the Swedish courts employ the 'free sifting/consideration of evidence (*fri bevisprovning*)' which, in principle, means that there are 'no limitations concerning the sources that can be used by the parties in a trial' (p. 2). Nevertheless, until recently, the direction of the law has been in support of the idea that Sweden is a homogeneous society which has meant that foreigners have been seen as 'inferior' and that they have suffered discrimination as a result of this attitude. Rabo discusses, in some detail, cases litigated by the Roma and the Sami against the state for discrimination in which these ethnic minorities have had to find their own experts and have fought long legal battles against the state, the police, private companies and individuals. With regard to the 'new immigrants'—the largest group are Syrian refugees/immigrants—it is clear that there has been some difficulty in finding appropriate experts to give evidence and that while most have only submitted reports to a court, they have not been called to give oral testimony. Rabo concludes by noting that the term 'culture' is highly ambivalent because it has been increasingly used by populists/demagogues and 'mainstream opinion makers'. Because the Swedish state 'is quite blind to its multi-cultural history', she argues that it is up to anthropologists/cultural experts and more progressive lawyers to work together to pursue justice for minorities.

---

22  See: https://culturalexpertise.net/wp-content/uploads/2020/04/EURO-EXPERT-Italydatasummary.pdf.
23  See: https://culturalexpertise.net/wp-content/uploads/2020/06/Sweden.pdf.

The situation in Poland differs from the above cited cases. In Holden's survey[24] of judges, lawyers and experts, it was found that 58% of judges and 45% of lawyers had never instructed an expert to provide CE. When CE had been submitted, it was predominately in family law (15%), crime (14%), refugee and asylum law (14%) and in immigration (12%). Burdziej (2019a) provides a more in-depth study of CE in Poland. He identifies the need for CE to deal with growing cultural diversity facing the immigration services, detention centers and in education—to address issues relating to migrants—and in the military (to prepare Polish troops who are deployed overseas). He undertook a key word search of legal databases—which held four hundred eighty-eight thousand decisions by Poland's appellate courts, administrative courts, and the Supreme Court—which yielded four hundred cases in which some form of CE was submitted (p. 12). He found that in the rare cases which involved CE that the majority 'concerned aspects of Polish dominant culture, not intercultural issues' (p. 13). The principle cases identified were asylum cases, hate speech crimes, defamation cases, media ethics, copyright, and religion. Burdziej found a very small number of cases which turned on CE: namely an honor killing, an attempt to kill a man responsible for an immigrants' daughter's suicide and Islamic terrorism.[25] He thought that many cases which raised cultural issues were simply 'filtered' out of the legal system[26] because 'Poland's relative cultural homogeneity leads courts to perceive culture as a largely non-problematic issue, thus not requiring the assistance of experts' (p. 25).

Burdziej has written a second paper titled 'Judging the Communist Past' (Burdziej 2019b) in which he provides a fascinating analysis of the use of historical expertise submitted to Polish courts which hear cases involved in 'lustration' ('vetting') the right of individuals to hold public office who are accused of collaborating with the communist secret services', cases to change the names of public places named after prominent communists, the removal of Communist monuments and the 'withdrawal of veteran status' from soldiers of the communist security services. The evidence submitted by historians comes from their analysis of what is left of the *Stasi* (secret service) archives (many files were destroyed or 'privatized' for the purpose of blackmail). The work of this court has become highly politicized.

A special appellate court was established in 2006 as was a new law of lustration which defined key definitions of what acts amounted to collaboration. Between 2008 and 2015 the court issued 903 decisions: 459 individuals were found to have been collaborators. Observers have noted the excessive leniency of the court which has led Burdziej to conclude that 'the courts often prefer erring on the side of the defendant' (p. 6). Controversy surrounding the courts decisions have, in part, ensured that prosecutors who submit evidence are 'careful to limit their claims to factual statements on the contents of the (Stasi) archive and not their interpretation as evidence of collaboration' (p. 8). Burdziej argues that this historical expertise has become institutionalized by the state, through the creation of the Institute of National Remembrance, in an effort 'to secure victory in the struggles over the dominant interpretation of the country's past' (p. 22).

In Denmark, which has 'a particularly monocultural, legally homogeneous, monolithic' legal system, Vinding (2019) argues that legal culture has changed very slowly as a result of the impact of migration, religion and EU legislation. Vinding undertook a key word search of the Danish legal database 'Karnov' for the period 2001 to 2018 which held approximately thirty-six thousand cases. He was able to identify one hundred and eight cases in which CE was submitted. The cases primarily concerned the Immigration Service and the Police Intelligence Service, and they concerned expulsion, terrorism, the prosecution of hijackers and cases brought by Iraqi nationals against the Danish Defense

---

[24] See: https://culturalexpertise.net/wp-content/uploads/2020/04/Poland.pdf.

[25] Terrorist criminal trials, and Closed Material Proceedings, need to understand the statements and concepts relied upon by the accused to justify their actions including the notion of *jihad*, *Da'wah* and other Islamic concepts. However, courts tend to disregard evidence on social, cultural, and contextual issues (see J. R. Campbell 2020b).

[26] If a 'filtering out' is occurring, this would also involve the police and state/public prosecutors, and would suggest that an examination of judicial decisions alone is only able to provide a partial understanding of the legal process and the role of CE.

Forces. Vinding argues that there is a need for the Danish courts to accept CE, but also a considerable reluctance by the judiciary to abandon its traditional approach to deciding cases.

Vinding's conclusions are supported by the discussion which Vetters and Foblets (2016) had with European judges. They concluded that judges are heavily influenced by their 'legal culture' and 'working environment' which predisposes them to adopt a pragmatic understanding of the 'culture' of immigrants 'that is not so much built on a clear-cut distinction between 'our' and 'their' culture, but rather focuses on notions of culture more directly related to the legal and organizational context within which judges operate' (p. 273). Based on a limited look at asylum proceedings in three countries—the U.K., Belgium, and Germany—the authors argued that judges are constrained by evidentiary norms from introducing CE.

Finally, a study by J. R Campbell (forthcoming) analyzes the role of CE in the British asylum system—including the rules which govern the submission of evidence—in first instance asylum hearings and on appeal to the Upper Tribunal of the U.K.'s Immigration and Asylum Chamber. These judges rely on CE/expert evidence to decide cases. Of the twenty-eight precedent-setting 'country guidance' cases decided by the Upper Tribunal between 2015 and 2019, he analyzed five and found that Immigration Judges

> use procedural rules to hedge and control expert evidence and that their assessment of this material is often flawed. This is particularly the case with qualitative data, which is treated as *hearsay*, but problems also occur with their preference for, but inability to adequately understand/ 'test', statistical data.

Campbell concluded that Immigration Judges

> lack the training to adequately assess/test testimonial, qualitative and statistical evidence. The problem is that they do not realize their limitations and end up preferring evidence which they assume is more objective or scientific while setting aside cultural evidence on language, culture, kinship and the importance of social relations and social networks. Legal conceptions of objective evidence used by IJs . . . do not assist them to assess social science research . . .

Holden reports on a survey of judges, lawyers and experts in the United Kingdom.[27] She reports that the majority of judges have never instructed an expert (33%; whereas 61% had instructed 10 or less reports) whilst 47% of lawyers had instructed less than 10 cases. Her findings are highly problematic, given that judges are not supposed to instruct experts—though they might be involved in agreeing which experts should give evidence—and solicitors but not barristers should instruct experts.

Research on international criminal courts provides a different picture of the role of CE. Maucec (2020) has, via the examination of a small number of cases, examined the limited availability of expert evidence on 'cultural' issues—i.e., efforts to mount a 'cultural defence', the meaning and use of 'fetishes and mystical power' and 'speech crime'—that have arisen in the International Criminal Court (ICC). Once again, however, the writer's belief in the value of cultural expertise is belied by the limited extent to which CE appears to have been submitted to and accepted by the ICC.

Anders (2014, p. 427) has written about the Special Court for Sierra Leone and argued that the judges 'refused to adopt the anthropologists' arguments (against the evidence relied upon by the prosecution) and did not share their concerns about the methodology and conceptual framework employed by the prosecution experts.' Anders' argues that that anthropological evidence was rejected because the judges 'were reluctant to recognize the challenge posed by Sierra Leone's socio-cultural specificities to the application of international criminal law.' In sharp contrast, Wilson (2015) compiled and analyzed a database of 473 expert reports submitted to the International Criminal Court for

---

27 See: https://culturalexpertise.net/wp-content/uploads/2020/06/UnitedKingdom.pdf.

the former Yugoslavia. He found that while 'scientific experts' were called to give evidence twice as often as were other experts, in fact 'social researchers (who provided both quantitative and qualitative data) have a much higher citation rate' than medical, military and police, document verification, financial and engineering experts (p. 732). Judges welcomed socio-cultural and linguistic evidence; however, their receptiveness to expert evidence 'depends on how jealously judges protect their preeminence as the triers of fact' (p. 741). Wilson's findings are supported by Eltringham (2013) who studied the International Criminal Court for Rwanda. There are clearly a number of factors which influence the reception of CE in international criminal courts, which, in addition to the issues already discussed, include judicial conceptions about culture as something practiced by 'distant (read backward) communities', the court's failure to appreciate its own organizational culture (as somehow objective and situated above socio-cultural specificities) and the judges responsibility to assign culpability (Fraser and Leyh 2020).

The above studies provide insight into the diverse ways that courts in different countries and jurisdictions view, instruct, admit and assess CE/expert evidence; however, the studies rely on a small number of published cases and fail to engage directly with judges. In what follows, I seek to fill key gaps in the research cited above, which has not adequately assessed the views of lawyers and the judiciary regarding the value of CE, nor has research assessed the full extent to which CE/socio-legal expert evidence has been submitted in national legal systems.

## 4. How British Barristers Make Use of 'Experts'

To obtain a clearer picture of the role of CE in the law, I interviewed five London-based barristers who undertake work on behalf of claimants (not the government). The zoom-interviews which ranged from fifty to ninety minutes in length were recorded and transcribed into Word. I asked barristers to identify and discuss at least three legal cases which they were personally involved in which relied upon different types of expert evidence. I also asked barristers to provide me with relevant case material or summaries of their cases which were published on Bailii. Because the published cases identify my informants, I have drawn upon but not cited this material in order to ensure the anonymity of my informants. The barristers I interviewed worked in the following jurisdictions: asylum and immigration law, family law and public law.

The *Asylum and Appeal Act 1993*[28] made it possible for asylum claimants to appeal against the decision of an Immigration Officer to the Immigration and Asylum Tribunal; at roughly the same time, legal aid became available to pay lawyers to litigate these cases. The appeal procedures were initially 'informal' and lawyers relied on reports published by international human rights organizations or the U.S. Department of State; it was rare for an expert to be instructed to provide a report and unheard of for an expert to be cross examined. Over time the situation changed: argument in the Tribunal has become increasingly 'legalized' and 'experts' have increasingly been instructed to provide evidence. Rather than provide its own expert evidence, the Home Office criticizes the experts instructed by claimants and attempts to undermine their reports (Home Office 2005, p. vi). In addition, once lawyers were appointed as Immigration Judges, it became increasingly common to cross-examine experts. This shift has meant that all forms of evidence are much more closely scrutinized in a process which has seen experts who rely on qualitative research and those whose reports are based on the analysis of documents in the public domain, i.e., reports by international organizations, to be deemed by IJs as insufficiently credible. Barristers argue that an analysis of country guidance decisions shows that IJs prefer experts who can analyze published statistics on casualties, death rates, risk of injury, mental health, and so on for a given population rather than experts who possess a good knowledge of an asylum applicant's country of origin.

---

[28] See: https://www.legislation.gov.uk/ukpga/1993/23/contents.

Barristers rely upon the solicitor who instructs them in a case to identify and instruct relevant experts. Solicitors also set out the Terms of Reference which experts are required to address; barristers may amend the ToRs but they are seldom in contact with experts. Barristers who litigate immigration and asylum claims rely heavily on 'country' experts—anthropologists, sociologists, historians, linguists and journalists who provide CE, as well as experts on foreign law—whose claim to expert knowledge rests on having conducted academic research on the country from which an asylum applicant originates. To identify an expert, immigration solicitors rely on recommendations from other practitioners and/or they contact experts listed on an online directory of country experts.[29] Because some barristers are skeptical about the individuals listed on the directory, they undertake research to identify a suitable expert, they read case law, google university webpages and consult colleagues in their chamber. Once identified and instructed, experts vary in their ability to fully address the terms of reference, in terms of their knowledge of the specific issues raised by a case, and in terms of their willingness to revise their reports to address barristers' concerns before their report is submitted to the court. Barristers who identify experts for their cases are less likely to end up relying on a person who produces generic material and/or who 'recycles' material which they have submitted in other cases (such experts tend to carry little weight with IJs).

Apart from selecting an expert who can address the specific issues raised in a case, a barrister's litigation strategy reflects a number of factors. First, all barristers had very clear views regarding how judges sitting in the jurisdiction in which they worked were likely to assess evidence. Thus, all the barristers' I spoke with held negative views regarding how IJs in the Immigration and Asylum Tribunal assessed expert evidence. One barrister thought that senior IJs were unwilling to take decisions which might be understood by the Government as 'impeding the operation of immigration control.' For this reason, barristers suggested that IJs failed to adequately reconcile international legal frameworks with British law in case their decisions challenged government policy. In order to 'square the circle' without overturning case law, it was suggested that IJs diminished the weight attached to expert evidence.[30] Barristers expressed a much more positive view of the quality of judge craft in jurisdictions where judges had been appointed to a judicial post following a successful legal career, i.e., the Family Division, the High Court, the Court of Appeal and so on.

Barristers who litigated across different jurisdictions provided further insights. For instance, appeals in the Family Court are often linked to claims in the Immigration and Asylum Tribunal which creates a 'gordian knot'—i.e., the need to ensure that both courts consider the decision of the other court—which can result in delays and problems for appellants. This situation arises in: deportation cases which raise an Art. 8 claim relating to the appellant's right to family life or the Human Rights claims of a child whose parent is to be deported; in care or adoption proceedings in the Family Courts; and with respect to claims regarding the threat of FGM to children threatened with deportation (Clark-Platts n.d.).[31] A barrister working in both courts stated that in the Family Courts—unlike in immigration proceedings—both parties must agree on a single expert to provide a report and that judges in the Family Court 'are much more likely to make positive findings from that evidence' and their findings 'tends to undercut adverse findings by IJs 'who criticize expert evidence and who 'think and act like border guards'. The barrister also noted that in the Family Court, legal counsel for the Secretary of State tend to adopt 'a more muted role' in proceedings.

---

[29] See: https://www.ein.org.uk/experts (accessed on 20 September 2020).

[30] See J. R. Campbell (forthcoming). The extent to which experts are attacked and their evidence undermined can be found in a number of appeals, e.g., *HH & others (Mogadishu: armed conflict: risk)* Somalia CG [2008] UKAIT *00022* where the Tribunal attacked and discredited several country experts who had provided cultural evidence. In *KV (Sri Lanka) (Appellant) v Secretary of State for the Home Department* (Respondent) [2019] UKSC *10* the Supreme Court overturned the Tribunal's approach to expert medical evidence.

[31] See: 'Deportation: interrelationship between family and immigration proceedings affecting the best interests of children', UK Immigration Justice Watch Blog at: https://ukimmigrationjusticewatch.com/2014/09/19/deportation-inter-relationship-between-family-and-immigration-proceedings-affecting-the-best-interests-of-the-children/ (accessed on 15 August 2020).

Interestingly, some barristers chose not to instruct experts particularly in the IAT when IJs focus on expert evidence and may fail to appreciate the wider picture. This decision reflected their strategy in mounting a judicial review against a government department, or their view of relevant case law in which experts had been 'trashed' by IJs. For example, one barrister relied on evidence submitted by IT experts in previous 'test cases' in the IAT to successfully mount a judicial review. The evidence—which was available online and was published by a Government Committee—concerned a notorious Home Office decision to set aside the examination results of fifty-thousand overseas students in the U.K. who were attending British educational institutions and who were required to pass the 'Test of English for International Communication' (TOEIC). The Home Office had wrongly decided that student test results were falsified and it revoked the right of overseas students to remain in the U.K. and required them to leave without completing their studies (House of Commons 2019).

In a different case, a barrister acting on behalf of an intervenor in a case heard by the Supreme Court, adduced unpublished social science research to provide the court with 'a macro-perspective' regarding the Government's failure to decide policy without taking into consideration the impact of its decision on sections of the British public. In this case, the barrister decided not to call an expert to submit evidence or testify. A third case involved a judicial review against a 'Conclusive Grounds' decision by the Home Office which had refused to recognize a young woman as a victim of trafficking. While the barrister could have instructed a country expert, she chose instead to instruct an ex-police officer who had extensive knowledge of trafficking in southern Europe. The official provided extensive evidence about corruption in the border force of the applicant's country of origin and the limited efforts by the Home Office to secure intelligence from Europol or Interpol to verify the evidence it relied on. The expert evidence, together with written evidence provided by organizations supporting victims of trafficking, successfully challenged the Home Office policy to refuse to reconsider negative trafficking decisions. All three cases were successful, and all relied on very different types of expert evidence or on legal argument.

It should not be too surprising, given the need to identify specific types of experts to address distinctive legal or factual issues in cases heard in different legal jurisdictions, that a wide range of experts are instructed and that, apart from immigration and asylum proceedings, cultural expertise seems to play a limited role. Nevertheless, interviews with five barristers provide insight into their use of experts, but a limited basis on which to generalize.

## 5. Cultural Evidence in British Law: A Survey of Bailii

How widely is CE used in British law? If it is used, to what extent and in what legal jurisdictions is it used in? To address gaps in the research cited above—and keeping in mind Golan's (2008) conclusion that 'scientists' have been providing expert evidence in British courts since at least the 18th century—I undertook a key word search of decisions made by all seven appellate courts and the Supreme Court in England and Wales to identify all types of expert evidence involved in legal cases.[32] The cases are held on Bailii (the British and Irish Legal Information Institute) <https://www.bailii.org/> a legal data base created in 1999. The database states that some, but not all, legal decisions are entered onto it and that 'Bailii makes available on the Internet a collection of leading cases identified by the legal academic community to support legal education' (Quick Guide to Bailii 2020). Given the long history of the British courts, it is best to assume that Bailli does not contain all the legal cases that have been decided in England and Wales. For this paper, I surveyed all the cases listed on the database in 2019—which contained nearly forty one thousand cases—by undertaking a key word search using the words 'expert', 'expert evidence', 'culture' and 'cultural' across all legal jurisdictions. I then read the cases identified by the key word search to identify the expert evidence submitted in each case.

---

[32]    The term 'expert' is commonly used in British case law. As a reviewer of this paper suggested, the word derives from the Latin term 'expertus' which was initially used to refer to any kind of 'consultant' called in to the court to give evidence.

Case decisions reported on Bailii are appeals from first instance courts; even when expert evidence was submitted, the reported cases on Bailii do not fully summarize the case or the evidence initially submitted. If the appeal concerns an issue of fact, then a decision briefly summarizes the evidence offered by the expert, the police and so on. However, many appeals are concerned with issues of law, not issues of fact, which means that expert evidence was not submitted. As will become clear, the British appellate courts hear cases involving individuals, firms and so on who are based in and outside the U.K.

*The Family Division of the High Court* hears cases where a child who is the subject of legal proceedings must be protected and this protection is not possible under the *Children Act 1989*. The most common type of case is where a child is made a 'ward of the court'.[33] It can also hear cases about forced marriage, Female Genital Mutilation[34], applications for financial relief where a divorce has taken place outside England and Wales and cases involving parental access to children.[35] The Family Court will normally hear all other cases about family issues, but may transfer some cases to the High Court if complex issues are involved.

In 2019, this court decided sixty-one cases. A key word search for the terms expert evidence/expert identified twenty-nine cases (forty-eight percent of all cases) in which the term was used. I found that the following types of experts were cited: expert accountant/forensic accountant, expert social worker, DNA/blood test evidence and handwriting experts.[36] A key word search for the terms culture/cultural identified thirteen cases (twenty-three percent). In these cases, testimony was provided by social workers, the police, and social services. In each case the term was used in a generic sense as in the culture of one of the parties in the case, cultural heritage, cultural needs (of children in care), cultural practices or cultural forms of abuse (i.e., by a foreign parent).

*The Court of Protection* makes decisions on financial or welfare matters for people who 'lack mental capacity' to make these decisions. The court is based in London where most cases are heard by District Judges and a senior judge. In 2019, this court decided fifty-eight cases. A key word search for the terms expert evidence/expert identified thirty-seven cases (sixty-four percent of all cases). I read four (eleven percent) cases and identified the following types of experts[37] who submitted evidence: medical experts, social workers, nursing experts and experts on foreign law. A key word search for the term culture/cultural identified ten cases (fourteen percent) and I read two cases (twenty percent). No expert evidence was provided in the first case; in the second case psychiatrists and social workers provided

---

[33] This court also handles cases of international child abduction but only if the abduction falls under either *The Hague Convention on the Civil Aspects of International Child Abduction* or *Brussels II Regulation (EC) No. 2201/2003*.

[34] Once evidence of FGM has been found by a doctor/nurse, the case is reported to the police for prosecution. Given the extent to which particular ethnic/minority groups are targeted as likely to have their children 'cut', it is surprising there is no requirement for CE to be submitted to the court. In the only case that has been successfully prosecuted in the UK, there was evidence of 'witchcraft'. See: 'UK. First successful prosecution of FGM' at: https://www.loc.gov/law/foreign-news/article/united-kingdom-first-successful-prosecution-for-female-genital-mutilation/. See Fontein (2014).

[35] For example, in 2017 I provided CE/expert evidence in a case in the Family Court regarding parental access to the children of a divorced couple. During proceedings the High Court Judge mediated between the two parties to arrive at an arrangement that was acceptable to both parents and which safeguarded the interests of the children.

[36] Ministry of Justice (2013, pp. 11–12) guidelines for experts make it clear that socio-legal experts are unlikely to be involved in the family courts. The Guidelines stipulate that experts should have: a 'working knowledge of the social, developmental, cultural norms and accepted legal principles applicable to the case presented at initial enquiry, and has the cultural competence skills to deal with the circumstances of the case'; that 'professional practice is regulated by a UK statutory body'. Furthermore, '[I]f the expert's area of professional practice is not subject to statutory registration (e.g., child psychotherapy, systemic family therapy, mediation, and experts in exclusively academic appointments) the expert would be expected to demonstrate appropriate qualifications and/or registration with a relevant professional body on a case by case basis. Registering bodies usually provide a code of conduct and professional standards.'

[37] These are civil claims. The rules of admissibility state that '[T]he question of admissibility was held to turn on four considerations: (i) whether the proposed expert evidence would assist the court in its task; (ii) whether the witness has the necessary knowledge and experience; (iii) whether the witness is impartial in his or her presentation and assessment of the evidence; and (iv) whether there is a reliable body of knowledge or experience to underpin the expert's evidence.' See Ministry of Justice (2016).

evidence. The term 'culture' was used in a generic sense as in 'cultural grounds' (i.e., the orientation of one of the parties) or the 'cultural norms' of one of the parties.

The *England and Wales High Court (Commercial Division)* hears complex national and international business disputes.[38] Many of these cases can also be heard by the Circuit Commercial Court. However, it generally hears the more complex cases, or cases where there is a large amount at stake. This court decided three hundred and thirty cases in 2019. A key word search for the terms expert evidence/expert identified a total of one hundred and seventy-three cases (fifty-two percent). A careful reading of ten of these cases (nearly six percent) showed that the following types of experts had submitted evidence: on ship sale and purchase (ship valuations), foreign law, valuation of costs, on ship cargo, arbitration, handwriting and on commercial transactions. A key word search for the terms culture/cultural identified twenty cases. I read three cases (fifteen percent) which indicated a generic use of the term 'culture' as in 'bank culture' (in dealing with cases of fraud) and the 'cultural fit' of a manager in a bank.

The *Chancery Division of the High Court* deals with: (a) disputes relating to business, property or land; (b) disputes over trusts; (c) competition claims under European or U.K. competition law; (d) commercial disputes (domestic and international); (e) intellectual property issues; and (f) disputes over the validity of a will ('probate disputes').[39] This court decided seven hundred and two cases in 2019. A key word search for the terms expert/expert evidence identified two hundred and sixty-nine cases (thirty-eight percent of the total). I read twenty cases (seven percent) which indicated that the following types of experts were involved: accountancy, liquidators, and experts in foreign law. However, there was a much larger use of the term generically as in: witness evidence, claimant's evidence, oral evidence, documentary evidence, lack of/no evidence and 'evidenced in writing'. With regard to a key word search for the terms culture/cultural, a total of thirty-eight cases (just over five percent) used the term. I read four cases (ten percent) and all four used the term in a generic sense, as in 'litigation culture', 'cultural activities', the culture of a business and 'company culture'. No expert evidence was adduced in these cases.

The *Court of Appeal Civil Division* hears appeals against certain decisions from all three divisions of the High Court of Justice and their specialist courts, including the Administrative Court: the county courts and the Family Court. It also hears appeals against certain decisions by other Tribunals.[40] Of particular relevance to this paper, this court hears appeals from the Upper Tribunal of the Immigration and Asylum Chamber which relies on a wide range of experts including anthropologists, historians, linguists and others in deciding asylum and immigration appeals[41] and age-dispute claims relating to assessing the age of child asylum seekers.[42] This court also decides Judicial Review applications.

This court decided four hundred and seventy-seven appeals in 2019. A key word search using the term expert evidence/expert identified a total of two hundred and nine (forty-four percent) cases. I read ten of these cases (five percent) which identified the following types of expert who submitted evidence: child psychologist, social workers, orthopedic experts, medical evidence, transport experts, coroners, and experts in foreign law. A key word search for the terms 'culture'/cultural expert identified

---

[38] The cases included: (a) disputes over contracts and business documents; (b) insurance and reinsurance; (c) the sale of commodities; (d) import, export and transport ('carriage') of goods; (e) issues relating to arbitration awards; (f) banking and financial services; (g) agency and management agreements; and (h) construction of ships.

[39] It also hears appeals about: (a) decisions of 'masters'; (b) insolvency decisions made by High Court registrars or the County Court; (c) most decisions of the County Court; and (d) decisions of certain tribunals. It can also handle a wide range of other issues including: (a) claims relating to partnerships (e.g. dissolution); (b) cancelling, setting aside or correcting ('rectifying') errors in deeds and other legal instruments; (c) breaches of trust or contract and (d) professional negligence.

[40] Including the Competition Appeal Tribunal; Employment Appeal Tribunal; Upper Tribunal (Administrative Appeals Chamber); Upper Tribunal (Immigration and Asylum Chamber); Upper Tribunal (Lands Chamber) and the Upper Tribunal (Tax and Chancery Chamber).

[41] A total of one hundred and nine cases were appealed to the CoA from the Immigration and Asylum Chamber in 2019 (sixteen percent of all the appeals heard by the CoA). These cases focus on 'errors of law' made by the Upper Tribunal.

[42] Age-dispute litigation has been heard by the Court of Appeal, the Supreme Court, and the High Court. In 2011 these cases were transferred to the Upper Tribunal of the Immigration and Asylum Tribunal. These cases require 'expert evidence' from paediatrician's, social workers, medical doctors, carers, foster parents, and teachers (see J. R. Campbell 2020c).

seventy-eight cases (sixteen percent). A reading of eight cases (sixteen percent) revealed that in only one case was evidence adduced, namely by a medical expert; I found that the term was used generically, e.g., cultural ties, cultural needs, culturally appropriate placement, cultural reasons, a child's culture and 'culturally integrated'.

*The Court of Appeal (Administrative Division)* reviews decisions made by people or bodies with a public law function, e.g., local authorities, and regulatory bodies. It hears judicial review applications made by other courts, tribunals and public bodies and it hears challenges to decisions made by certain people or bodies (e.g., ministers or local government) where legislation provides a right to challenge.[43] It also contains a specialist Planning Court which handles judicial reviews of decisions about planning permission and challenges to planning decisions. It is a specialist court within the Queen's Bench Division of the High Court of Justice, which is based at the Royal Courts of Justice, London and in Birmingham, Cardiff, Leeds, and Manchester. Cases may be heard by one High Court judge or by a 'Divisional Court' which consists of two or more judges, normally a High Court Judge and a Lord Justice of Appeal.

In 2019, this court decided six-hundred and eighty-two appeals. A key word search for the terms expert/expert evidence identified two hundred and sixty-one cases (thirty-eight percent of all cases). A careful reading of seventeen cases (six percent) identified the following types of experts who submitted evidence: medical, foreign law, firearms, planning, prison/independent psychologist, experts on human trafficking, the Environmental Agency, parole boards, planning, Migration Advisory Council expertise, Nature England, government regulatory experts, veterinary experts, midwives and experts on public health. A key word search for the term 'culture'/cultural identified ninety cases (thirteen percent of the total). A reading of nine cases (ten percent) revealed that in three cases expert evidence was submitted by a human rights monitor, an archaeologist, a psychiatrist and a psychologist. The term was also used in a generic sense, e.g., sub-culture, cultural norms, cultural heritage, cultural identity, cultural differences, cultural consideration, compliance culture and culture of benefit dependency.

*The Court of Appeal (Criminal Division)* hears appeals from the Crown Courts. It hears appeals against convictions and sentences (even if the conviction was in a magistrate's court) and confiscation orders imposed by the Crown Court. It also heard applications for permission to appeal and appeals from proceedings in the Crown Court (including cases referred by the Attorney General where there is concern that the sentence given by the Crown Court may have been too lenient).[44] It is based at the Royal Courts of Justice in London. Cases are heard by Lord Justices of Appeal or, in some cases, High Court judges.

In 2019 this court decided four hundred and six appeals. A key word search for the terms expert/expert evidence identified seventy-seven cases (nineteen percent). I read seven of these cases (nine percent) and identified the following types of experts who submitted evidence: the analysis of drugs, health and safety experts, psychiatrists, firearms experts, and medical experts. A key word search of cases for the terms culture/cultural identified fourteen cases (three percent). A careful reading of two cases (fourteen percent) found that no expert evidence was submitted and that the term was used generically, e.g., culture of secrecy and gang culture.

*England and Wales Queen's Bench Division of the High Court* hears disputes relating to personal injury; negligence; breach of contract; breach of a statutory duty; breach of *The Human Rights Act 1998*;

---

43  This court also hears: (a) applications for 'habeas corpus', (a legal procedure where the court rules on whether the detention of an individual is legal); (b) applications to prevent individuals from continuing to initiate groundless legal proceedings (i.e. 'vexatious litigants') from continuing to do so without first obtaining permission from a court; (c) all applications under the *Coroners Act 1988* (which deals with the appointment and conduct of coroners); (d) appeals 'by way of case stated' from the Crown Court or magistrates' courts (where an opinion is sought on a particular point of law where a mistake may have been made): (e) applications for an order to imprison a person for contempt of court; (f) appeals under the *Extradition Act 2003* (which deals with extradition requests to and from the United Kingdom and decision on bail applications); (g) appeals against decisions made by some professional bodies, e.g. the Nursing and Midwifery Council; and (h) applications for 'restraint orders' or 'certificates of inadequacy' where assets have been frozen or confiscated.

44  In addition, it hears appeals from decisions from the 'Court Martial Appeal Court'.

libel, slander and other torts; and non-payment of a debt and 'enforcement orders. Many of these cases can also be heard by the Chancery Division.[45] In 2019, this court decided five hundred and seventeen appeals. A key word search for the terms expert/expert evidence identified two hundred and forty-two cases (forty-seven percent). I read fifteen cases (six percent) which identified the following types of expert[46] who submitted evidence: on road accidents, handwriting, medical, neurologist, on the price of drugs, on general medical practice, diabetology, foreign law and clinical psychology. A key word search for the terms culture/cultural identified forty-five cases (nine percent). A careful reading of four cases (nine percent) failed to identify a case in which expert evidence was involved and found only a generic use of the term, as in 'cultural reasons', culture of the defendant, culture (of a foreign country) and institutional culture.

Finally, I reviewed decisions by the United Kingdom's *Supreme Court* which is the final court of appeal in the U.K. for civil and criminal cases from England, Wales, and Northern Ireland. It hears cases of the greatest public or constitutional importance. In 2019 the Supreme Court decided fifty-nine appeals. A key word search for the terms expert/expert evidence identified twenty-two cases (thirty-seven percent). I read four cases (eighteen percent) which identified three types of experts who submitted evidence: medical, handwriting, and judicial experts. A key word search for the terms culture/cultural identified a total of eight cases (thirteen percent); my reading of three cases (thirty-seven percent) found that references were made concerning the expertise of the Department for Digital, Culture, Media and Sport (a government department), and to cultural life and cultural integration (in relation to a deportation appeal).

A lengthy history of immigration has led the U.K. government and British courts to recognize and enshrine certain aspects of the culture of ethnic minorities into British law, e.g., with respect to the right of Sikhs to wear a turban, a 'kara' or a 'kurban'[47], the right of Muslim women to wear a burqa[48], and limited rights accorded to Roma peoples.[49] Legislation has also been adopted which bars ethnic and religious discrimination.[50] Civil law cases in the U.K., which in the past may have called upon anthropologists/socio-legal experts to provide evidence on matters of culture, family, gender, religion and marriage, are now dealt with by the police who enforce civil rights legislation by imposing civil penalties rather than prosecuting individuals in court.

In keeping with studies of the role of CE in other countries discussed above, in England and Wales, the majority of the appellate court decisions recorded on Bailii do not involve CE[51] or expert evidence of any kind (because they are concerned with deciding/interpreting points of law not factual issues). Even so, some form of expert evidence is submitted in all the appellate courts in the U.K. because they

---

[45] This Court also handles: applications to 'enroll' (register) deeds, including changing your name by deed poll; registration of judgments obtained abroad so that they can be enforced in England and Wales; election petitions to challenge the results of Parliamentary, European Parliamentary and local government elections; applications for bail; serving documents overseas and obtaining evidence for foreign courts; registration and satisfaction of 'bills of sale'; and 'interpleader proceedings' where a High Court Enforcement Officer is attempting to recover goods to settle a debt and a third party claims to be the owner of the goods.

[46] Experts are regulated by Part 35 of the Civil Procedure Rules. However, the Queen's Bench Guide (Ministry of Justice 2018, sec. 10.8) requires parties to obtain permission from the court before they secure an expert, and both parties are given a very strong steer that they should agree a single, joint expert. The Guide also makes it clear that, 'The most common form of written evidence is a witness statement' (sec. 10.9.2).

[47] See: Grillo (2017) on litigation which expanded the right not to be discriminated against as set out in the *Equalities Act 2010* and the *Human Rights Act 1998*.

[48] See endnote 36, and the discussion on 'face coverings' in 'What are the rules on burqas and face coverings in the UK' at: https://fullfact.org/law/what-are-rules-burkas-and-niqabs-uk/ (accessed on 20 August 2020).

[49] The rights of Roma are not protected by legislation in the UK, though litigation has secured a measure of protection against discrimination (see: Willers and Baldwin n.d.).

[50] These rights are set out in the *Equalities Act 2010*.

[51] The limited extent to which expert evidence is cited in appellate decisions does not indicate whether doctors, psychologists, experts on human trafficking provide 'socio-cultural'/CE evidence. Given the strict regulations on the submission expert evidence in British courts which require an expert to have an acknowledged 'expertise' in their area of work, it seems unlikely that other professionals are allowed to provide CE that is in any way analogous or similar to anthropologists, historians and other social scientists.

hear appeals from first tier courts. The survey of Bailii indicates that the term 'expert' is understood in a very wide sense to include nearly every type of expertise that arises in a modern, technologically complex society.[52] It is notable that none of the cases identified by a key word search involved anthropological/CE for several reasons. First, anthropological/CE evidence, when it is submitted, is heard by the lower courts which are deemed to possess greater expertise in dealing with evidence. Secondly, most first instance cases are not appealed to the appellate courts. Third, it is probably the case that a key word search of a database like Bailii does not identify many relevant cases because the search engine is unable to identify the issues relating to CE/socio-legal expertise. This, in turn, probably means that it is not possible to determine the extent to which the courts deal with CE and whether, over time, courts have accepted more or less CE/expert evidence. Finally, and equally importantly, a key word search of Bailii together with an analysis of the cases requires substantially more resources and time than individual researchers have at their disposal to undertake a comprehensive analysis of the database. Ideally, what is required are more in-depth studies of countries and legal jurisdictions by teams of researchers (supported by external funding). Such studies would enable us to move beyond the analysis of individual cases, as useful as these are for illuminating judicial decision-making, and surveys of legal databases to provide a more complete understanding the role of all legal actors—the police, state prosecutors, lawyers, experts and judges. It is important to know how decisions are made to prosecute or drop cases, e.g., by filtering out and preventing claims from reaching court and, once a claim reaches court, to understand the work of all legal actors involved in providing evidence and arguing and deciding claims in which CE is submitted.

## 6. Conclusions

An assessment of the contribution of cultural expertise in assisting judges or mediators to better understand the cultural issues which arise in disputes has proven to be a complicated task. A review of the burgeoning literature reveals several issues. First, anthropologists and a wide range of socio-legal experts have increasingly submitted CE/expert evidence in asylum, civil, criminal, family law, indigenous land claims and in claims heard by international criminal courts. It is clear that these experts have quite different reasons for engaging with the courts, ranging from the desire to protect subaltern peoples to profiting from their work. Many, perhaps most, have a limited understanding of the adversarial process.

Second, regardless of jurisdiction and country, judges act as gatekeepers (a fact that has not been recognized in many studies). They exercise this role by imposing evidentiary hurdles and admissibility rules, in their assessment of the evidence, and by deciding legal claims. The judiciaries power is only slightly mitigated by the discretion given the lower courts—particularly asylum and immigration hearings and in civil claims—to admit hearsay/opinion evidence which, occasionally, can have a decisive impact on a judicial decision.

Third, courts in different countries and in different jurisdictions appear to exercise considerable discretion with regard to admitting socio-legal/CE evidence. This probably arises from the extent to which they are insulated from, or are required to engage with, international humanitarian and criminal law. The failure of courts in Denmark, Poland and Sweden seem to indicate that their failure to engage with international law is due in part to their concern to maintain a 'mono-cultural' society. The insular focus of these courts may also explain why so little CE is submitted and thus why judges and lawyers are relatively unfamiliar with CE/socio-legal expertise.

An appreciation of the extent to which CE is submitted in different jurisdictions and countries is constrained by the limitations of existing research which has primarily been based on the analysis of a limited number of judicial decisions. Recently, researchers have attempted to provide a much wider

---

[52] The role of certain of social workers, psychiatrists and lawyer/mediators in arbitration has been discussed by Brophy et al. (2013), Hallett (2018) and De Girolamo (2013), respectively.

picture of CE by searching legal databases. These studies are problematic for two reasons. The first type of studies are illuminating but they tell us relatively little about the work of all the legal actors—the police, public prosecutors, lawyers, experts and judges—whose work shapes the adversarial process and determines the role and value of CE. Surveys, on the other hand, hold out the promise of situating CE/socio-legal expertise in the wider adversarial system, but findings provide at best a limited picture due to methodological and technical problems involved in searching databases.

Finally, a greater appreciation of how the law works shows that the key differences between anthropology and law are not epistemological but reflect the need for the court to reach a decision. While experts are clearly regulated by the judiciary, it should be clear that the situation they face varies considerably by country and by jurisdiction and that the principal use of CE is in asylum and immigration law, civil claims, international criminal law and secondarily in international human rights law and family law.

**Funding:** This research received no external funding.

**Acknowledgments:** I am grateful to the five barristers who agreed to be interviewed about their litigation strategies and to the three anonymous reviewers for their comments on the initial version of this paper.

**Conflicts of Interest:** The author declares no conflict of interest.

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
