# Peer review of "Interrogating the Role and Value of Cultural Expertise in Law"

_laws, 2020_

Round 1

Reviewer 1 Report

I found this article very interesting but confusing. I will start were I will conclude this is one of the rare articles that I wish the author has written more, provided more examples and facts to support his or her contentions.  I will state that I am trained as a lawyer and not as an anthropologist so I may not have understood some of the nuance in the article’s contentions. If so, I apologize.

First, I was left a bit unsure of the article’s domain. The abstract seemed to focus on cultural experts, but the text seemed to narrow this concept to anthropologists-but in other places, the articles seem to broaden the scope of the cultural expert. I think that the article would be stronger if this term was more clearly defined and then used consistently. As I read, this article, I wondered if sociologists or psychologists were excluded. They appeared to be at least implicitly by the choice of language in the text. Also, in the end the focus of the article seems to be on immigration and criminal cases and in one jurisdiction. Lines 64—66. This is a significant limitation that the scope of the original research is British law during a narrow one year window of time (2019) with no real explanation as to why this year was chosen, how representative is it of the broader body cases, where there any significant intervening changes in law that may affect the value of this data stet. See generally lines 343-571. The author would in discussing the author’s analysis state I found x cases but only read y cases with no explanation as to how those cases were chosen. But, the initial claim and language throughout the article seems to imply a broader claim or at least one that should be more generalizable. I think that this should have been set up better introduction. I was able to follow the article because I started with the abstract.

The article generalizes to the point of being stereotypic at least too much in the US context on concepts of hearsay, the use therefore by experts, and their role at trial. I think that all of the practical assumptions of how courts work, how triers of fact work (judge or jury), would have been strengthen had the author looked at the law review literature, especially by authors of treatises written for practicing lawyers. In terms of how expert witness testimony is used by juries, I believe that there is a robust body of literature written by jury consultants, and at least there should have been some discussion of jury instructions given by the judges as a practical explanation of how things work. The author may choose to reject these sources but if so, the reader deserves an explanation.

There are places that I was left unsure if the citation supported the claim, for example footnote 2, line 50. I am not sure that US Federal Rules of Evidence 704 sets aside usurping the ultimate issue. Further, the claim is US and other countries, but the citation is only to a US rule of evidence. Also, that claim refers to guilt (the article in places conflates criminal and civil standards).

In line 113, the article states may challenge the law, I am not sure that this means in the context it is presented. In line 117, there is the conflation on litigant/defendant with the criminal standard of guilty/not guilty. There is much more going on here, especially in the context of civil proceedings that may involve rights or liberties. Line 135, I am not sure that the ‘ is needed in the word individuals. Lines 141-143 refer to transitional justice. I am not sure if this term is being used differently but I usually think of “Transitional justice refers to the ways countries emerging from periods of conflict and repression address large-scale or systematic human rights violations so numerous and so serious that the normal justice system will not be able to provide an adequate response.” See https://www.ictj.org/about/transitional-justice#:~:text=Transitional%20justice%20refers%20to%20the,to%20provide%20an%20adequate%20response..

This meaning does not seem to fit into this context as written. The discussion of the role of the expert and the expert’s politics does not seem to reflect the relatively narrow purpose of the expert in a legal proceeding to assist the trier of fact to reach a decision nor the legal and ethical obligation of the expert to speak to truth based on the norms of their discipline using the tools of their discipline. See (US) Federal Rules of Evidence 704. This is another place where looking at the words of judges and practices may have been helpful.

Lines 195-196, again I am curious how does the expert anthropologist challenge judiciary.

Lines 198-204, I think a good discussion of Daubert (US) or Gilfoyle or O'Doherty (UK) in terms of what is admissible as expert testimony.

Lines 243-44, footnote 15 I would like to know more about this case.

The description of the research methodology needs to be more detailed and explain better the hypothesis getting into the research and why things were included or excluded and any limitations. See 331-342.

This article has definite potential. It just needs to be developed more. This is the rare case when the author should have written more. Rather than conclusory statements and citations incorporate the facts from the reference or examples into this paper. Also, the audience is unclear, is this article an argument to lawyers and judges, solicitors or barristers, or to anthropologists. As a lawyer, I would like to learn more about what anthropologists have to offer and what is keep their value added out of the fact-finding process. I worry and the author does not present evidence to the contrary that the reason this expert testimony is not entering is because of the appropriate gatekeeping function of the court’s under Daubert. Court are is out experts who are opining with a sound foundation on topics that they could not get past a peer reviewed journal, if their testimony was a publication. “The Daubert standard requires the court to act as a gatekeeper, allowing the admission of an expert opinion only if the court determines that the methodology supporting the expert opinion is sufficiently reliable by considering:

  • Whether the methodology or theory can be or has been tested, peer reviewed, or published.
  • The known or potential rate of error for the methodology or technique.
  • The existence and maintenance of standards controlling the operation of the methodology.
  • The degree to which the methodology or theory is generally accepted in the relevant scientific community.”

Author Response

Reviewer #1

I found this article very interesting but confusing. I will start were I will conclude this is one of the rare articles that I wish the author has written more, provided more examples and facts to support his or her contentions.  I will state that I am trained as a lawyer and not as an anthropologist so I may not have understood some of the nuance in the article’s contentions. If so, I apologize.

First, I was left a bit unsure of the article’s domain. The abstract seemed to focus on cultural experts, but the text seemed to narrow this concept to anthropologists-but in other places, the articles seem to broaden the scope of the cultural expert. I think that the article would be stronger if this term was more clearly defined and then used consistently. As I read, this article, I wondered if sociologists or psychologists were excluded. They appeared to be at least implicitly by the choice of language in the text. Also, in the end the focus of the article seems to be on immigration and criminal cases and in one jurisdiction. Lines 64—66. This is a significant limitation that the scope of the original research is British law during a narrow one year window of time (2019) with no real explanation as to why this year was chosen, how representative is it of the broader body cases, where there any significant intervening changes in law that may affect the value of this data stet. See generally lines 343-571. The author would in discussing the author’s analysis state I found x cases but only read y cases with no explanation as to how those cases were chosen. But, the initial claim and language throughout the article seems to imply a broader claim or at least one that should be more generalizable. I think that this should have been set up better introduction. I was able to follow the article because I started with the abstract.

[Thanks for your comments. I have yellow highlighted all changes to the paper in the resubmission. As a result of comments from all three reviewers, the article is now longer than the original version. I have also changed the abstract to reflect a somewhat different and more nuanced focus which is to explore research on the role of socio-legal/cultural expertise in the courts of western Europe and North America. As a result of a broader discussion, I have flagged up more clearly the relevance of CE in civil claims, asylum & immigration law, international criminal law, family law and, to a lesser extent, criminal law. The paper explains why I restricted my search of Bailii to 2019: in that year 40,000 cases were decided. There are simply too many cases to identify and examine given my resources; the idea was to explore the limits of research on databases and to set out in some detail the nature of expert evidence in British courts.]

The article generalizes to the point of being stereotypic at least too much in the US context on concepts of hearsay, the use therefore by experts, and their role at trial. I think that all of the practical assumptions of how courts work, how triers of fact work (judge or jury), would have been strengthen had the author looked at the law review literature, especially by authors of treatises written for practicing lawyers. In terms of how expert witness testimony is used by juries, I believe that there is a robust body of literature written by jury consultants, and at least there should have been some discussion of jury instructions given by the judges as a practical explanation of how things work. The author may choose to reject these sources but if so, the reader deserves an explanation.

[The paper now looks at hearsay/opinion evidence in the US and the UK; there is no evidence that a judges instruction to the jury or jury selection plays an important role in these case. Indeed, as now discussed in the paper, the importance of jury trial in north America and the UK has substantially declined; and juries play a very limited role on the continent; see my discussion of this on p. 11-12]

There are places that I was left unsure if the citation supported the claim, for example footnote 2, line 50. I am not sure that US Federal Rules of Evidence 704 sets aside usurping the ultimate issue. Further, the claim is US and other countries, but the citation is only to a US rule of evidence. Also, that claim refers to guilt (the article in places conflates criminal and civil standards).

[see my comments above]

In line 113, the article states may challenge the law, I am not sure that this means in the context it is presented. In line 117, there is the conflation on litigant/defendant with the criminal standard of guilty/not guilty. There is much more going on here, especially in the context of civil proceedings that may involve rights or liberties. Line 135, I am not sure that the ‘ is needed in the word individuals. Lines 141-143 refer to transitional justice. I am not sure if this term is being used differently but I usually think of “Transitional justice refers to the ways countries emerging from periods of conflict and repression address large-scale or systematic human rights violations so numerous and so serious that the normal justice system will not be able to provide an adequate response.” See https://www.ictj.org/about/transitional-justice#:~:text=Transitional%20justice%20refers%20to%20the,to%20provide%20an%20adequate%20response.

[Some of the cited authors indicate cases where expert/CE evidence overturned existing law, see Rosen 1977; I have added additional cases. Line 135 is a quote. I agree with your comments on Transitional Justice, please see my comments in the text]

This meaning does not seem to fit into this context as written. The discussion of the role of the expert and the expert’s politics does not seem to reflect the relatively narrow purpose of the expert in a legal proceeding to assist the trier of fact to reach a decision nor the legal and ethical obligation of the expert to speak to truth based on the norms of their discipline using the tools of their discipline. See (US) Federal Rules of Evidence 704. This is another place where looking at the words of judges and practices may have been helpful.

[While the law may wish to regulate expert evidence, I hope it is clear from the research I cite that many experts seek to challenge the law even as they assist the trier of facts]

Lines 195-196, again I am curious how does the expert anthropologist challenge judiciary.

[I have given some examples; see Rosen quote at beginning of paper on civil claims]

Lines 198-204, I think a good discussion of Daubert (US) or Gilfoyle or O'Doherty (UK) in terms of what is admissible as expert testimony.

Lines 243-44, footnote 15 I would like to know more about this case.

[I am citing Jones 1994]

The description of the research methodology needs to be more detailed and explain better the hypothesis getting into the research and why things were included or excluded and any limitations. See 331-342.

[see highlighted text at p. 14]

This article has definite potential. It just needs to be developed more. This is the rare case when the author should have written more. Rather than conclusory statements and citations incorporate the facts from the reference or examples into this paper. Also, the audience is unclear, is this article an argument to lawyers and judges, solicitors or barristers, or to anthropologists. As a lawyer, I would like to learn more about what anthropologists have to offer and what is keep their value added out of the fact-finding process. I worry and the author does not present evidence to the contrary that the reason this expert testimony is not entering is because of the appropriate gatekeeping function of the court’s under Daubert. Court are is out experts who are opining with a sound foundation on topics that they could not get past a peer reviewed journal, if their testimony was a publication. “The Daubert standard requires the court to act as a gatekeeper, allowing the admission of an expert opinion only if the court determines that the methodology supporting the expert opinion is sufficiently reliable by considering:

  • Whether the methodology or theory can be or has been tested, peer reviewed, or published.
  • The known or potential rate of error for the methodology or technique.
  • The existence and maintenance of standards controlling the operation of the methodology.
  • The degree to which the methodology or theory is generally accepted in the relevant scientific community.”

[I have now written more and provided an expanded discussion of some of the literature. I have found no evidence of CE used in criminal trials in the US (which is not to say that it doesn’t happen]. For this reason a further discussion of Daubert is not relevant for the cases discussed in the paper. Many of the cases which are cited are judge-led with no jury]

Reviewer 2 Report

Great review to the literature on anthropological contribution of CE in litigation. Below are some minor editing suggestions/questions. Additions are indicated in bold; questions in [….]:

line 85: South Asians (Holden 2011; Menski 2013[Cap "south"; missing ")"]

line 91: hearsay [Check font of hearsay throughout paper: it has appeared as hyphenated in line 323]

119: on (judges quickly act to rule...[Why "(" before judges....]

168: "The final issue...." could follow the previous sentence in line 167 without a need for a new paragraph.

170 formatting issue for sentence

321-322, 325-325: formatting issue

329, 330: IJ [First time use IJ in article; spell out IJ; clean up …; …]  

616, 618:  Burdziej 2019; 2019a [Check order of publication dates]

645:Cooke, T. [missing publication year]

700: Jones, C. [missing publication year]

Author Response

Reviewer #2

Great review to the literature on anthropological contribution of CE in litigation. Below are some minor editing suggestions/questions. Additions are indicated in bold; questions in [….]:

line 85: South Asians (Holden 2011; Menski 2013[Cap "south"; missing ")"]

line 91: hearsay [Check font of hearsay throughout paper: it has appeared as hyphenated in line 323]

119: on (judges quickly act to rule...[Why "(" before judges....]

168: "The final issue...." could follow the previous sentence in line 167 without a need for a new paragraph.

170 formatting issue for sentence

321-322, 325-325: formatting issue

329, 330: IJ [First time use IJ in article; spell out IJ; clean up …; …]  

616, 618:  Burdziej 2019; 2019a [Check order of publication dates]

645:Cooke, T. [missing publication year]

700: Jones, C. [missing publication year]

[Thanks for your comments. I have made all the editing suggestions you identified]

Reviewer 3 Report

The article addresses a crucial subject in the current socio-legal debate and it does this with the intent to provide an assessment of cultural expertise (CE) in legal proceedings. At this aim, the author proposes a twofold analysis, one more general (the perspective of experts and the perspective of judges on CE) based on the existing literature, and one more specific analysis, based on the author’s own research: a survey conducted by the author on a UK database Bailii.

The structure of the paper is interesting and promising. The comparison between the view of the experts, on one side, and judges on the other side, is useful to show the different perspectives. This is of interest for readers. As well the Bailii survey is interesting to enhance knowledge about CE in the UK.

While the structure is solid and some results original and worthy to be brought to the attention of the public, there are some issues in developing the argument and assessing CE that need, in the opinion of the reviewer, to be addressed to warrant publication.

The main issues concern:

  • the connection between the first general part of the paper (16-342) and the second part concerning the survey (343-571). For instance, in the first part strong general statements are made that seem to refer to CE in all legal proceedings, while in part two the focus is on UK. This is not an issue per se, but several times the paper draws general conclusions that are not true for all the states in Europe. It seems, sometimes, that the declared intent in the title and the abstract is too ambitious for what than is proved within the paper (see below for specific suggestions).
  • methodological criticism about existing literature is not accompanied by solid evidence. The paper took strong position of criticism with all existing scholarship on CE (cultural expertise). Those criticisms are questionable as they seem to imply that only quantitative research is acceptable in this field. The reader understands that only “quantitative methodology” is a true source of research in this field, which is highly questionable (see below for details).

The following changes are recommended to improve the paper (please note: numbers are the lines where the correction is suggested; sentence in " " are parts of the paper):

5 In the sentence “and which is primarily the result of qualitative research” it is not clear to what “which” refers: to CE, to information or to this paper? It might be clearer to eliminate the sentence or to create a separate sentence.

6 The sentence “to understand how effective and useful CE has been” looks to be too ambitious once read the paper. It would be better to circumscribe the scope of the analysis, for example by adding “has been” “in Western literature and in UK law” or “in the UK” or “in North America and Europe”. This depend on how the author wants to enhance the connection between part I and II of the paper and solve the doubt between different level of assessment of CE (in all existing legal proceedings, in North America and Europe or in UK).

10-11 This final part of the abstract does not match with the conclusion that rather stress other points (e.g. the fact that the UK judges use a broader type of experts) please align abstract with the text or vice versa.

17 introduce references on scholarship referred to at the beginning of line 16.

31 “in this paper I assess the role of CE in judicial proceedings”: this sentence is too ambitious and too broad for what the rest of the paper analyses. Please add a sentence like, “as emerging in Western literature and in UK law” after judicial proceedings to clarify the extension of the analysis.

62 please clarify for not UK readers that Bailii stands for British and Irish Legal Information Institute. Please, correct the typo: BAILLI, in BAILII. There are several typos like this all along the paper: please change Bailli in Bailii everywhere.

65-66 “namely asylum and immigration law”: this conclusion may be true for UK but not for other States where CE plays a role in criminal law, family law, and for national minorities or Roma people. It would be better in all the paper not to talk in general of CE, but rather specify when the statement regards Europe, North America or UK. Another way in which the author may improve this issue is to nuance the strong statements so far made in the paper as conclusive truths, by inserting adverbs like: generally, often, in some States.

85 there is a typo: please, close parenthesis after “Menski 2013”

117-118 “judges have little patience with arguments about cultural relativity”. This sentence is contestable: there are several case-law, both in Europe and in North America that show that judges embrace the point of view of the defendant supported by CE even if is radically different from the culture of the judge. It would be important to introduce some nuances in this strong statement.

331-342 The criticism outlined here is weak and problematic.

On one side, it is inconsistent to tell that “each of the studies cited above is flawed” as at line 217 the work of Jones has been defined as “excellent”.

On the other side, the reasons given to criticize the scholarship are not scientifically sound. For instance, the fact of being “anecdotal and incomplete” to me is not a flaw. At the contrary, it is inherent to the fact that the methodology used by scholarship to analyse the subject of CE is often based on case-law. It should be taken into account that even the analysis of a single case-law and of the role played by the CE there can be significant. By making this criticism, the author seems to imply that only quantitative research counts in this field, while this is very disputable. In fact, excellent results have been produced by papers analysing just one case-law in which CE was performed.

If the author is truly convinced that only research on database and a quantitative approach are valid to assess CE, this should be proved with more arguments.

I personally recommend not to adopt such a controversial stance with other existing literature, as the results of this paper move toward a different direction, and they can be integrated with existing literature.

339 “and other types of expert”: it would be useful to add “other types of socio-legal expert evidence”. In fact other types of expert is very broad.

343 ff. the word Bailii is sometimes written in the wrong acronym Bailli: there are several typos of this kind along all the paper, please find them and change in Bailii. It would be useful to clarify in the title of the section that Bailii stand for British and Irish Legal Information Institute. I recommend to insert it in parenthesis like that: “A survey of Bailii (British and Irish Legal Information Institute)” in line 343

481-2 “The survey of Bailii indicates that the term expert is understood in a very wide sense to include nearly every type of expertise that arises in a modern… society”. It is not clear why this fact should be relevant for the purpose of the paper to assess CE. The multiple use of the term expert seems to be more a problem of lexicon. English and other languages use the law expert for every kind of expertise. This homogeneity comes from Roman law where the word “peritus” (expert) was used for any kind of consultancy the judge required. For the scope of this paper I do not think it is relevant to say that the word expert is the same.

Instead, I think it would be important to better stress what emerge form the analysis of the Bailii, that is that there is a sort of “competition” between experts called by the judges. This is an interesting point the analysis shows.

It would be interesting also tell if the experts which are doctors, psychologists, experts on human trafficking can act as “cultural brokers” (thus enhancing the concept of CE as suggested by Livia Holden) and if they bring cultural knowledge into the trial, although they are not anthropologists. This point it is not clear, and, as reader, I felt the curiosity to know more about this: did the survey find some result in this sense?

573 Please, correct typo: As assessment, in An assessment

578 Please, correct typo: ini

587 the criticism of existing research as anecdotal is not methodologically sound. To the reader, this implies that only quantitative research in the field of CE is valid, while, at the contrary, research based on single case-law has been very important to evaluate the role of cultural experts in the legal process.

Author Response

Reviewer #3

The article addresses a crucial subject in the current socio-legal debate and it does this with the intent to provide an assessment of cultural expertise (CE) in legal proceedings. At this aim, the author proposes a twofold analysis, one more general (the perspective of experts and the perspective of judges on CE) based on the existing literature, and one more specific analysis, based on the author’s own research: a survey conducted by the author on a UK database Bailii.

The structure of the paper is interesting and promising. The comparison between the view of the experts, on one side, and judges on the other side, is useful to show the different perspectives. This is of interest for readers. As well the Bailii survey is interesting to enhance knowledge about CE in the UK.

[Thanks for your comments. I agree with your commens on the structure of the paper and I have restructured the paper to discuss my material on British barristers before my survey of Bailii and to make the argument flow more coherently. Note that the focus of the paper is now on western Europe and North America]

While the structure is solid and some results original and worthy to be brought to the attention of the public, there are some issues in developing the argument and assessing CE that need, in the opinion of the reviewer, to be addressed to warrant publication.

 The main issues concern:

  • the connection between the first general part of the paper (16-342) and the second part concerning the survey (343-571). For instance, in the first part strong general statements are made that seem to refer to CE in all legal proceedings, while in part two the focus is on UK. This is not an issue per se, but several times the paper draws general conclusions that are not true for all the states in Europe. It seems, sometimes, that the declared intent in the title and the abstract is too ambitious for what than is proved within the paper (see below for specific suggestions).

[I agree and have revised the structure and provided further information on Europe on p. 11-12]

  • methodological criticism about existing literature is not accompanied by solid evidence. The paper took strong position of criticism with all existing scholarship on CE (cultural expertise). Those criticisms are questionable as they seem to imply that only quantitative research is acceptable in this field. The reader understands that only “quantitative methodology” is a true source of research in this field, which is highly questionable (see below for details).

[I have rewritten this – my focus is on qualitative not quantitative research]

 The following changes are recommended to improve the paper (please note: numbers are the lines where the correction is suggested; sentence in " " are parts of the paper):

 5 In the sentence “and which is primarily the result of qualitative research” it is not clear to what “which” refers: to CE, to information or to this paper? It might be clearer to eliminate the sentence or to create a separate sentence.

[changed]

 6 The sentence “to understand how effective and useful CE has been” looks to be too ambitious once read the paper. It would be better to circumscribe the scope of the analysis, for example by adding “has been” “in Western literature and in UK law” or “in the UK” or “in North America and Europe”. This depend on how the author wants to enhance the connection between part I and II of the paper and solve the doubt between different level of assessment of CE (in all existing legal proceedings, in North America and Europe or in UK).

[I changed the text and the abstract]

 10-11 This final part of the abstract does not match with the conclusion that rather stress other points (e.g. the fact that the UK judges use a broader type of experts) please align abstract with the text or vice versa.

[as above]

 17 introduce references on scholarship referred to at the beginning of line 16.

[there are too many; suggestion is not practical]

 31 “in this paper I assess the role of CE in judicial proceedings”: this sentence is too ambitious and too broad for what the rest of the paper analyses. Please add a sentence like, “as emerging in Western literature and in UK law” after judicial proceedings to clarify the extension of the analysis.

[done]

 62 please clarify for not UK readers that Bailii stands for British and Irish Legal Information Institute. Please, correct the typo: BAILLI, in BAILII. There are several typos like this all along the paper: please change Bailli in Bailii everywhere.

[done]

 65-66 “namely asylum and immigration law”: this conclusion may be true for UK but not for other States where CE plays a role in criminal law, family law, and for national minorities or Roma people. It would be better in all the paper not to talk in general of CE, but rather specify when the statement regards Europe, North America or UK. Another way in which the author may improve this issue is to nuance the strong statements so far made in the paper as conclusive truths, by inserting adverbs like: generally, often, in some States.

[I accept this criticism and have modified the text accordingly. Thanks]

 85 there is a typo: please, close parenthesis after “Menski 2013”

[done]

 117-118 “judges have little patience with arguments about cultural relativity”. This sentence is contestable: there are several case-law, both in Europe and in North America that show that judges embrace the point of view of the defendant supported by CE even if is radically different from the culture of the judge. It would be important to introduce some nuances in this strong statement.

[I accept this statement and have tried to do this in the revised text and also by expanding my analysis of some of the material – see the yellow highlighted text in the resubmission]

 331-342 The criticism outlined here is weak and problematic. On one side, it is inconsistent to tell that “each of the studies cited above is flawed” as at line 217 the work of Jones has been defined as “excellent”. On the other side, the reasons given to criticize the scholarship are not scientifically sound. For instance, the fact of being “anecdotal and incomplete” to me is not a flaw. At the contrary, it is inherent to the fact that the methodology used by scholarship to analyse the subject of CE is often based on case-law. It should be taken into account that even the analysis of a single case-law and of the role played by the CE there can be significant. By making this criticism, the author seems to imply that only quantitative research counts in this field, while this is very disputable. In fact, excellent results have been produced by papers analysing just one case-law in which CE was performed.

[I have rephrased the text and expanded my discussion accordingly. See. P. 15.

If the author is truly convinced that only research on database and a quantitative approach are valid to assess CE, this should be proved with more arguments. I personally recommend not to adopt such a controversial stance with other existing literature, as the results of this paper move toward a different direction, and they can be integrated with existing literature.

[perhaps you misunderstood me? I have rephrased my approach to the different studies]

 339 “and other types of expert”: it would be useful to add “other types of socio-legal expert evidence”. In fact other types of expert is very broad.

[done]

 343 ff. the word Bailii is sometimes written in the wrong acronym Bailli: there are several typos of this kind along all the paper, please find them and change in Bailii. It would be useful to clarify in the title of the section that Bailii stand for British and Irish Legal Information Institute. I recommend to insert it in parenthesis like that: “A survey of Bailii (British and Irish Legal Information Institute)” in line 343

[done]

 481-2 “The survey of Bailii indicates that the term expert is understood in a very wide sense to include nearly every type of expertise that arises in a modern… society”. It is not clear why this fact should be relevant for the purpose of the paper to assess CE. The multiple use of the term expert seems to be more a problem of lexicon. English and other languages use the law expert for every kind of expertise. This homogeneity comes from Roman law where the word “peritus” (expert) was used for any kind of consultancy the judge required. For the scope of this paper I do not think it is relevant to say that the word expert is the same.

[see endnote 31 and associated text]

Instead, I think it would be important to better stress what emerge form the analysis of the Bailii, that is that there is a sort of “competition” between experts called by the judges. This is an interesting point the analysis shows.

It would be interesting also tell if the experts which are doctors, psychologists, experts on human trafficking can act as “cultural brokers” (thus enhancing the concept of CE as suggested by Livia Holden) and if they bring cultural knowledge into the trial, although they are not anthropologists. This point it is not clear, and, as reader, I felt the curiosity to know more about this: did the survey find some result in this sense?

[they cannot; see endnotes 6 & 34 etc]

 573 Please, correct typo: As assessment, in An assessment

[done]

Round 2

Reviewer 1 Report

The article is interesting, and I think that it makes it point. A times it a bit overly ambitious in that a more narrow focused article on one legal system (common law vs civil law) or one country or region to developed its points might have been more persuasive.  That being said, the author makes good points, and I hope that he or she will in the future develop these ideas further. Good truly interdisciplinary work is very hard to find.

Author Response

Thank you for your valuable comments, I will in the future develop these ideas further